# FunSpace: A functional and spatial analytic approach to cell imaging data using entropy measures

**Thao Vu** *, **Souvik Seal, Tusharkanti Ghosh, Mansooreh Ahmadian, Julia Wrobel**¤, **Debashis Ghosh**

Department of Biostatistics and Informatics, University of Colorado Anschutz Medical Campus, Aurora, Colorado, United States of America

¤ Current address: Department of Biostatistics and Bioinformatics, Rollins School of Public Health, Emory University, Atlanta, Georgia, United States of America

* thao.3.vu@cuanschutz.edu

## Abstract

Spatial heterogeneity in the tumor microenvironment (TME) plays a critical role in gaining insights into tumor development and progression. Conventional metrics typically capture the spatial differential between TME cellular patterns by either exploring the cell distributions in a pairwise fashion or aggregating the heterogeneity across multiple cell distributions without considering the spatial contribution. As such, none of the existing approaches has fully accounted for the simultaneous heterogeneity caused by both cellular diversity and spatial configurations of multiple cell categories. In this article, we propose an approach to leverage spatial entropy measures at multiple distance ranges to account for the spatial heterogeneity across different cellular organizations. Functional principal component analysis (FPCA) is applied to estimate FPC scores which are then served as predictors in a Cox regression model to investigate the impact of spatial heterogeneity in the TME on survival outcome, potentially adjusting for other confounders. Using a non-small cell lung cancer dataset (n = 153) as a case study, we found that the spatial heterogeneity in the TME cellular composition of CD14+ cells, CD19+ B cells, CD4+ and CD8+ T cells, and CK+ tumor cells, had a significant non-zero effect on the overall survival (p = 0.027). Furthermore, using a publicly available multiplexed ion beam imaging (MIBI) triple-negative breast cancer dataset (n = 33), our proposed method identified a significant impact of cellular interactions between tumor and immune cells on the overall survival (p = 0.046). In simulation studies under different spatial configurations, the proposed method demonstrated a high predictive power by accounting for both clinical effect and the impact of spatial heterogeneity.

## Author summary

Studying spatial heterogeneity in the tumor microenvironment (TME) provides insights into tumor development and progression. In this work, we proposed a novel approach to account for the simultaneous heterogeneity caused by both cellular diversity and spatial

**Data Availability Statement:** The TNBC dataset is publicly available at https://www.angelolab.com/mibi-data. The NSCLC and ovarian cancer datasets are freely available as part of the Bioconductor

ExperimentHub package VectraPolarisData (https://bioconductor.org/packages/release/data/experiment/html/VectraPolarisData.html). The immunohistochemistry (IHC) data on lung tissues during COVID-19 progression can be found at https://doi.org/10.5281/zenodo.4633905. Open-source code and reproducible analysis script can be accessed at https://github.com/thaovu1/FunSpace.

**Funding:** T.V. is funded by the Grohne-Stepp Endowed Chair for Cancer Research from the University of Colorado Cancer Center. The funders had no roles in study design, data collection and analysis, decision to publish, or preparation of the manuscript.

**Competing interests:** The authors have declared that no competing interests exist.

organizations of multiple cell types, by leveraging spatial entropy measures. Functional principal component analysis (FPCA) was applied to subject-specific spatial entropy measures calculated at different distance ranges to extract FPC scores which captured the spatial heterogeneity in the TME cellular composition. By applying the proposed approach to the multiplex immunohistochemistry (mIHC) data of patients with non-small cell lung cancer (NSCLC) and a publicly available multiplexed ion beam imaging (MIBI) triple-negative breast cancer (TNBC) dataset, we explored the significant impact of the spatial heterogeneity in the TME cellular composition on patient overall survival while adjusting for other confounders.

## 1 Introduction

The emergence of tumor microenvironment (TME) studies has revealed a critical role of spatial heterogeneity for gaining insights into tumor initiation, development, progression, invasion, metastasis, and response to therapies [1–4]. The TME is known to be complex and heterogeneous due to continuous cellular and molecular adaptions in the primary tumor and its surroundings, which then allow for tumor growth and proliferation. Increasing evidence suggests that in addition to quantities and types, the spatial organizations of cells within the TME influences survival and response to treatment therapy in numerous cancer types [4]. For instance, [3] discovered a high level of heterogeneity in the TME of patients with human lung adenocarcinoma (LUAD) by comparing different tumor sites. Particularly, the authors identified the immunological differences in cell subpopulations between the core, middle, and edge of tumors, such that CD4+ naive T cells located at the core of the tumor had higher activation levels in angiogenesis and expressed more immune checkpoint molecules than those at the tumor edge. In another study about human papillomavirus (HPV)-negative head and neck squamous cell carcinoma (HNSCC) tumors, [4] demonstrated that neoplastic tumor-immune cell spatial compartmentalization, rather than mixing, was associated with longer progression free survival (PFS).

Significant advancements have been achieved in understanding the spatial organization of cells in the tumor microenvironment (TME) thanks to the development of advanced single-cell multiplex imaging techniques [5–10]. These innovative methods enable the simultaneous quantification and visualization of individual cells within tissue sections. More specifically, multiplex tissue imaging (MTI) [11] methods such as cyclic immunoflourescence (CyCIF) [12], CO-Dectection by indEXing (CODEX) [13], multiplex immunohistochemistry (mIHC) [8, 14], imaging mass cytometry (IMC) [15], and multiplex ion beam imaging (MIBI) [16, 17] have enabled the simultaneous measurements of of tens of markers at single-cell resolution while preserving the spatial distribution of cells. For instance, multiplex immunohistochemistry (mIHC) utilizes antibody-antigen reactions coupled to fluorescent dyes or enzymes to detect and visualize specific antigens in tissue sections [18, 19]. Alternatively, multiplexed ion beam imaging (MIBI) [16] takes a different approach by employing secondary ion mass spectrometry to image metal-conjugated antibodies. As a result, MIBI enables single-cell analysis of up to 100 parameters without spectral overlap between channels. Altogether, imaging provides an additional dimension of spatial resolution to the single-cell signature profiles, which in turn allows researchers not only to study cellular composition but also to make inferences about specific cell-cell interactions.

Metrics that quantify the spatial differences between TME cellular patterns can range from simple density ratios of immune cells to tumor cells within specific tumor regions (e.g., tumor

center vs. invasive margin) such as Immunoscore [20], to more complex measures utilizing spatial proximity of specific cell types relative to others in the TME such as mixing scores [5] and cellular neighborhood measures [7]. Alternatively, the Ripley's K-function and its variants could also be employed to characterize any single-cell spatial patterns deviated from the complete randomness at any given distance. However, for a given point pattern of multiple cell types, such spatial summary functions typically operate in a pairwise fashion, which involves all possible comparisons of one cell type to another, but not all types simultaneously.

Shannon entropy [21], initially proposed in Information Theory, can be used to measure the heterogeneity in observations. Its use has gained popularity in a wide range of applied sciences such as ecology and geography [22–24], evolutionary biology [25], landscapes [24, 26], and recently in cancer research [27, 28]. For instance, Heindl et al. [27] investigated the micro-environmental diversity of ovary tumors and the corresponding local metastasis sites including omentum, peritoneum, lymph node, and appendix by accounting for the collective characteristics of all cell types simultaneously via Shannon diversity index. Particularly, based on the cell type frequency distribution, a high Shannon score indicates similar proportion of each type while a low value suggests dominance of one cell type. In a different use case, Natrajan et al. [28] associated the diversity of cell composition in the tumor ecosystem with patients overall survival on a more granular level. Specifically, after cell phenotyping and classification, the authors divided each image into smaller regions such that Shannon entropy can be calculated for each region based on the frequency distribution of each cell type. Accordingly, a high value of the entropy indicate a heterogeneous environment while the reverse holds for a low entropy value. The distribution of Shannon diversity scores was then used as input for Gaussian mixture model to determine the number of clusters, which was referred to as an ecosystem diversity index (EDI). Recently, Wu et al. [29] quantified intra-tumor spatial heterogeneity in lung adenocarcinoma samples, termed geographic diversity (GD), using the Mantel correlation test [30] between the molecular information (e.g., protein expression levels) and geographic information (e.g., spatial locations of samples). Through a permutation test, the authors obtained the significance level of the observed Mantel correlations, which was then used to stratify patients into GD patterns: clustered (p-value < 0.01) vs. random (p-value > 0.01).

While the direct application of Shannon entropy in [27] shows promising results in ovarian cancer, it does not consider the spatial distribution of cell types. In other words, regardless of how cells of different types are distributed on a given image, the Shannon entropy is the same as long as the proportion of each cell type stays the same. The EDI approach, on the other hand, tries to overcome such challenge by considering small neighborhoods of cells through image tessellation. However, this approach relies heavily on how each image is tessellated (e.g., shape and size) and the chosen number of clusters from fitting the model to obtain the EDI score. Alternatively, our recent approach SPF [31] could be utilized to effectively model a nonlinear impact of the spatial interaction between any two cell types of interest on the overall survival. However, in the case of multiple cell types, the SPF approach needs to consider the spatial interactions between all pairs of different cell types to necessarily make inferences about such nonlinear impact.

Different from SPF, herein, we propose an approach that leverages spatial entropy measures [32] to account for the spatial heterogeneity of multiple cell types simultaneously across subjects at different distance ranges and how such variability impacts a clinical outcome of interest. More precisely, when incorporating space into entropy measures, the proposed framework is able to capture the diversity in cellular composition, such as similar proportions across cell types or dominance of a single type, at a specific distance range. Additionally, spatial patterns, including clustered, independent, or regular, among cell types can also be acquired. For instance, if cells of different types are randomly scattered on an image, the spatial entropy at

any given distance is close to zero (Fig 1A and 1C). On the other hand, the spatial entropy deviates from zero if spatial patterns of cell types in a given local neighborhood are different from the global pattern (Fig 1B and 1D). Depending on the distribution of distances across different cell types in each image, the distance ranges can vary across subjects, leading to irregularly observed subject-specific spatial entropy measures. Functional principal component analysis (FPCA) [33] which is known to be robust to irregular data is applied to the resulting spatial entropy measures to obtain FPC scores that capture the spatial heterogeneity of the TME cellular composition. The FPC scores are then used as predictors in a Cox regression model to investigate the association between spatial heterogeneity in cell composition with patient overall survival.

In this article, we applied the proposed approach to a non-small cell lung cancer (NSCLC) dataset, which was previously explored by Johnson et al. [34] regarding the pairwise spatial interactions of each immune cell type with cancer cells. The authors found that an increased

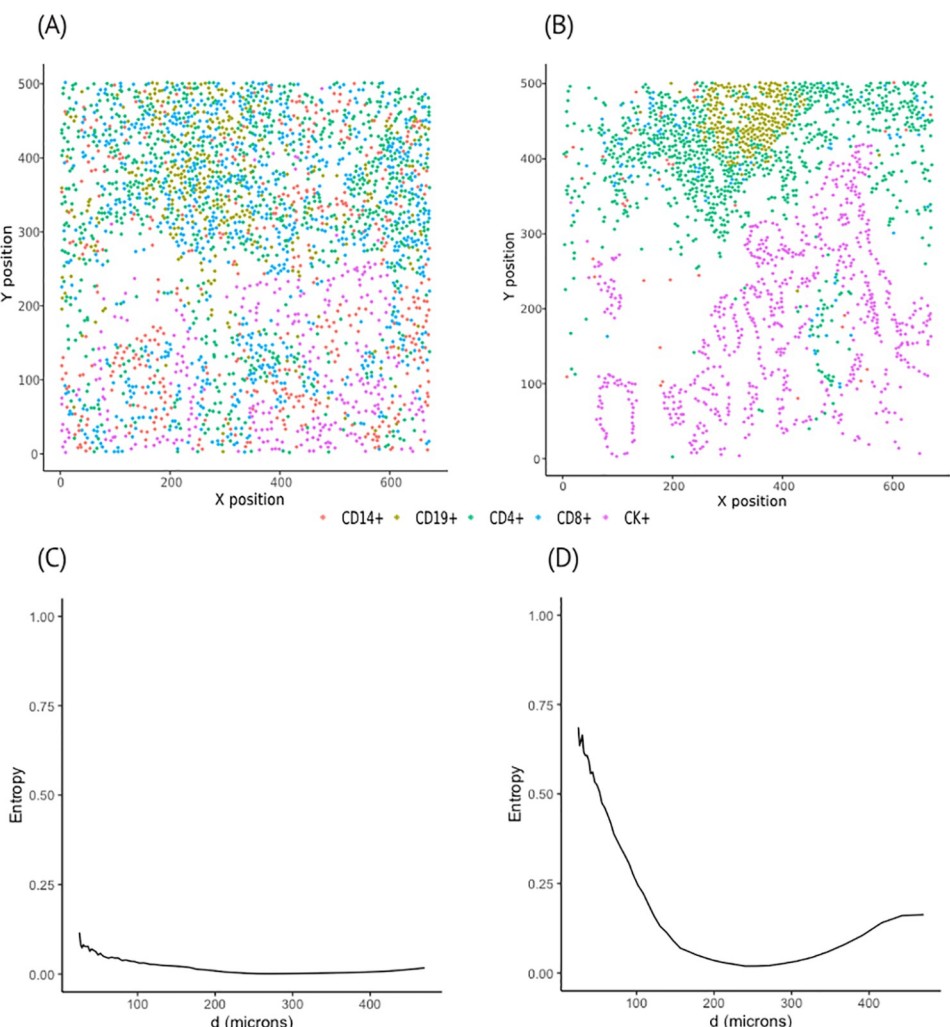

**Fig 1. Representative images and corresponding spatial entropy.** (A), (B): two representative images with individual cells of five different types: CD14+, CD19+, CD4+, CD8+, and CK+. (C), (D): corresponding spatial entropy curves at multiple distance ranges.

proximity between cancer cells to immune cells improved overall survival. Motivated by their findings, we simultaneously investigated the spatial heterogeneity in the TME cellular composition of CD14+ cells, CD19+ B cells, CD4+ and CD8+ T cells, and CK+ tumor cells, and their collective impact on the overall survival. Furthermore, using a publicly available multiplexed ion beam imaging (MIBI) triple-negative breast cancer (TNBC) dataset [5], we studied the impact of spatial interactions between tumor and immune cells on survival outcome. Finally, the predictive power of the proposed method was demonstrated with simulation studies considering different spatial configurations.

## 2 Materials and methods

### 2.1 Model

**2.1.1 Spatial entropy measures.** Let $Y$ be a discrete random variable denoting a cell type, with the total of $J$ possible types. Shannon entropy [21] is the expected value of an information function measuring the uncertainty in observing $Y = y_j$, $j = 1, \ldots, J$ with the corresponding probability mass function (pmf) as $p_Y = (p(y_1), p(y_2), \ldots p(y_J))^T$. The entropy is defined as:

$$H(Y) = \sum_{j=1}^{J} p(y_j) \log\left(\frac{1}{p(y_j)}\right) \tag{1}$$

The entropy $H(Y)$ ranges between 0 and $\log(J)$; and its maximum is achieved when $Y$ is uniformly distributed. In our context, the maximum entropy is reached when we have the same number of cells for all cell types in a given image. Note that the above entropy alone does not account for the role of space. As a result, datasets with identical pmf $p_Y$ but different spatial configurations (e.g., strong spatial association vs. complete spatial randomness) yield the same $H(Y)$.

Following [32], we define a new variable $Z$ to identify co-occurrences of different pairs of realizations of $Y$ over space, i.e. $(y_j, y'_j)$, with $j, j' = 1, \ldots J$. An assumption is that the order within co-occurrences is neglected. This is reasonable since the interest is to understand the spatial heterogeneity of data over a space, which is usually not assumed to have direction. The number of categories of $Z$, denoted by $R$, where $R = \begin{pmatrix} J+1 \\ 2 \end{pmatrix}$. The corresponding pmf is $p_Z = (p(z_1), p(z_2), \ldots p(z_R))^T$, where $p(z_r)$ for $r = 1, \ldots, R$, is the probability of observing the $r$th co-occurrence of cells on a given image. In other words, $Z$ transforms the information in $X$ while introducing a venue for incorporating the idea of spatial neighborhood. The entropy using the newly introduced variable $Z$ is defined as:

$$H(Z) = \sum_{r=1}^{R} p(z_r) \log\left(\frac{1}{p(z_r)}\right) \tag{2}$$

Here, the entropy $H(Z)$ ranges between 0 and $\log(R)$. Although the values of $H(Y)$ and $H(Z)$ might not be exactly the same due to different categories being considered, both capture equivalent information.

In order to properly account for space in an entropy measure, an additional random variable $W$ is defined to cover all possible distances at which co-occurrences take place. Denote $w_k = (d_{k-1}, d_k]$, with $k = 1, \ldots, K$ and $K$ is the total number of distance breaks. Here, $d_k$ is a fixed set of distance breaks which are a function of pairwise distances between all cells in the pattern, referred to as inter-cell distances. These distance breaks $d_k$ can be flexibly chosen depending on the specific applications. In our context, as the general focus was on local cell-to-cell interactions, we specifically set the initial distance $d_0$ to 0 and $d_K$ to be the median of the inter-cell

distance distribution. Then, $d_k$ can be generated as a sequence of distance breaks of length $K$ by linearly decreasing from $d_K$ to 0 on a log scale.

The corresponding pmf is denoted by $p(W) = (p(w_1), \ldots, p(w_K))^T$, with $p(w_k)$ being the probability of observing pairs of cells whose corresponding distances fall within the $k$th distance range. At each distance interval $w_k$, the probability of observing specific co-occurrences is represented as $p(Z|w_k) = (p(z_1|w_k), \ldots, p(z_R|w_k))^T$. The partial residual entropy after account for spatial information at each $w_k$, denoted by $H_k^W(Z|w_k)$, can be computed as follows.

$$H_k^W(Z|w_k) = \sum_{r=1}^{R} p(z_r|w_k) \log\left(\frac{1}{p(z_r|w_k)}\right), \text{for } k = 1, \ldots, K$$

Then, the overall residual entropy $H^W(Z)$ can be represented as a weighted summation of $H_k^W(Z|w_k)$.

$$H^W(Z) = \sum_{k=1}^{K} p(w_k) H_k^W(Z|w_k) \quad = \sum_{k=1}^{K} p(w_k) \sum_{r=1}^{R} p(z_r|w_k) \log\left(\frac{1}{p(z_r|w_k)}\right)$$

$$= \sum_{k=1}^{K} \sum_{r=1}^{R} p(z_r, w_k) \log\left(\frac{1}{p(z_r|w_k)}\right)$$

Similarly, the spatial information capturing the entropy due to space at each $w_k$, denoted as $SPI(Z|w_k)$, can be calculated as.

$$SPI(Z|w_k) = \sum_{r=1}^{R} p(z_r|w_k) \log\left(\frac{p(z_r|w_k)}{p(z_r)}\right) \text{for } k = 1, \ldots, K$$

The overall spatial information denoted by $SPI(Z)$, known as mutual information between $Z$ and $W$, can be computed as a weighted sum of $SPI(Z|w_k)$, as follows.

$$SPI(Z) = \sum_{k=1}^{K} p(w_k) SPI(Z|w_k) \quad = \sum_{k=1}^{K} p(w_k) \sum_{r=1}^{R} p(z_r|w_k) \log\left(\frac{p(z_r|w_k)}{p(z_r)}\right)$$

$$= \sum_{r=1}^{R} \sum_{k=1}^{K} p(z_r, w_k) \log\left(\frac{p(z_r|w_k)}{p(z_r)}\right)$$

$$= \sum_{r=1}^{R} \sum_{k=1}^{K} p(z_r, w_k) \log p(z_r|w_k) - \sum_{r=1}^{R} \log p(z_r) \sum_{k=1}^{K} p(z_r, w_k)$$

$$= -H^W(Z) - \sum_{r=1}^{R} p(z_r) \log p(z_r)$$

$$= H(Z) - H^W(Z)$$

In other words, by utilizing the two newly defined variables, $Z$ and $W$, we are able to decompose the global entropy $H(Z)$ in Eq 2 into two components: entropy due to space (i.e., spatial information) and the remaining heterogeneity after space has been taken into account (residual entropy), respectively denoted by $SPI(Z)$ and $H^W(Z)$, as follows.

$$H(Z) = SPI(Z) + H^W(Z)$$

Section S. 8 in S1 Text provides a detailed illustration for computing spatial and residual entropy. Given the additivity property, we can partition both $SPI(Z)$ and $H^W(Z)$ at each

distance range $w_k$, such that

$$SPI_k = SPI(Z|w_k) = \sum_{r=1}^{R} p(z_r|w_k) \log\left(\frac{p(z_r|w_k)}{p(z_r)}\right)$$

$$H_k^W(Z|w_k) = \sum_{r=1}^{R} p(z_r|w_k) \log\left(\frac{1}{p(z_r|w_k)}\right).$$

Since our interest lies in investigating the spatial heterogeneity across cellular configurations, we will focus on the spatial entropy $SPI_k$ for the rest of the paper. Fig 1 displays the distributions of five different cell types including CK+, CD4+ and CD8+ T cells, CD14+ cells, and CD19+ B cells in two representative images (A, B) and the corresponding spatial entropy $SPI_k$ as a function of inter-cell distances (C, D). Specifically, in the first spatial configuration Fig 1A, the influence of space is rather weak as all cell types seem to scatter evenly across all distance ranges Fig 1C. Spatial entropy values are relatively larger at short distances due to small clusters of CD4+ and CD19+ cells.

**2.1.2 Functional principal component analysis (FPCA).** Our goal is to use entropy measures defined in Section 2.1.1 for modeling patient-level outcomes. For each $i$-th subject, we have a collection of cells with their corresponding xy-coordinates available. We first define distance breaks and ranges for each $i$-th subject as $d_{ik}$ and $w_{ik}$, respectively, such that $w_{ik} = (d_{i(k-1)}, d_{ik}]$ for $k = 1, \ldots, K_i$ and $i = 1, \ldots, N$. Spatial entropy for the $i$-th subject is represented as $SPI_i = \{SPI_{ik}\}_{k=1}^{K_i} = \{SPI_i(Z|w_k)\}_{k=1}^{K_i}$. If the subject-specific number of distance breaks $K_i$ is too large, indefined entropy could occur at small distance ranges $w_{ik}$ with no point being observed. Thus, to ensure that spatial entropy can be computed for each $w_{ik}$, we limit $K_i \leq 50$. Section S.7.2 of S1 Text explored the impact of varying $K$ on the resulting spatial entropy curves. Since the distance ranges may vary across subjects depending on the relative distances between cells, the resulting spatial entropy functions $SPI_i$ can be irregularly observed. Modeling these SPI directly as functional covariates as in [31] is possible in principle but would require methodological developments that are beyond our scope. Instead, we perform functional principal components analysis (FPCA) to obtain functional principal component (FPC) scores that can be utilized as scalar covariates in a standard modeling framework [33].

Assume that spatial entropy for subject $i$ at distance break $k$, $SPI_{ik}$, is generated from the underlying smooth random function $X(s)$ at a random distance $s$, with known mean function $EX(s) = \mu(s)$ and covariance function $Cov(X(s), X(t)) = G(s, t)$. The domain of $X(.)$ typically is a bounded and closed distance interval $\mathcal{S}$. Suppose there is an orthogonal expansion of $G$ in terms of eigenfunctions $\phi_l$ and nonincreasing eigenvalues $\lambda_l$, such that $G(s, t) = \sum_l \lambda_l \phi_l(s)\phi_l(t)$ with $t, s \in \mathcal{S}$. The $i$-th random curve can be expressed as $X_i(s) = \mu(s) + \sum_l \xi_{il}\phi_l(s)$, $s \in \mathcal{S}$, where $\xi_{il}$ are uncorrelated random variables with mean 0 and variance $E\xi_{il}^2 = \lambda_l$ with $\sum_l \lambda_l < \infty$, $\lambda_1 \geq \lambda_2 \geq \ldots$. Let $SPI_{ik}$ be the $k$th observation of the random function $X_i(.)$ at a random distance $S_{ik}$, and $\epsilon_{ik}$ be iid measurement errors. The considering model becomes

$$
\begin{aligned}
SPI_{ik} &= X_i(S_{ik}) + \epsilon_{ik} \\
&= \mu(S_{ik}) + \sum_{l=1}^{\infty} \xi_{il}\phi_l(S_{ik}) + \epsilon_{ik}, \quad \text{for} \quad S_{ik} \in \mathcal{S}
\end{aligned}
\tag{3}
$$

where, $E\epsilon_{ik} = 0$, $var(\epsilon_{ik}) = \sigma^2$.

We implemented the R package fdapace by Yao et al. [33] to estimate the mean function $\hat{\mu}(s)$ and the covariance surface $\hat{G}(.)$ based on the pooled data from all subjects. Details of the

local linear smoothers for function and surface estimation are provided in Section S.1 of S1 Text.

Once the mean function and the eigenfunctions are computed, the FPC scores can be subsequently estimated. Due to limited data points $SPI_{ik}$ observed at discrete distances $S_{ik}$, the FPC scores can not be reasonably estimated using traditional numerical integration. Alternatively, FPC scores can be estimated through conditional expectation such that

$$\hat{\xi}_{il} = \hat{E}\{\xi_{il}|SPI_i\} = \hat{\lambda}_l \hat{\phi}_{il} \hat{\Sigma}_{SPI_i}(SPI_i - \hat{\mu}_i)$$

where the $(j, l)$-th element of $\hat{\Sigma}_{SPI_i}$ is $(\hat{\Sigma}_{SPI_i})_{j,l} = \hat{G}(X(S_{ij}), X(S_{il})) + \sigma^2 \delta_{jl}$ with $\delta_{jl} = 1$ if $j = l$, and 0 otherwise.

**2.1.3 Cox regression model.** The relationship between survival and the spatial heterogeneity embedded in the aforementioned FPC scores, in addition to clinical predictors $U_i = (u_{i1}, \ldots, u_{ip})^T$ can be investigated using a Cox regression model. Denote $T_i$ and $C_i$ as the survival and censoring times for the $i$-th individual, respectively. Assume that $T_i$ and $C_i$ are independent given $U_i$. Due to right-censoring, we only observe $Y_i = min(T_i, C_i)$. Let $\delta_i = I(T_i \leq C_i)$ be a censoring indicator. The hazard function for the Cox regression model has the form

$$
\begin{aligned}
\log h_i(t) &= \log h_0(t) + \sum_{j=1}^{p} u_{ij}\gamma_j + \sum_{l=1}^{\infty} \xi_{il}\beta_l \\
&\approx \log h_0(t) + \sum_{j=1}^{p} u_{ij}\gamma_j + \sum_{l=1}^{L} \xi_{il}\beta_l \\
&= \log h_0(t) + \mathbf{U}_i^T \boldsymbol{\gamma} + \xi_i^T \boldsymbol{\beta} \\
&= \log h_0(t) + \mathbf{W}_i^T \boldsymbol{\theta}
\end{aligned}
\tag{4}
$$

where $\mathbf{W}_i^T = (u_{i1}, \ldots, u_{ip}, \xi_{i1}, \ldots, \xi_{iL})^T$, $\log h_i(t)$ is the log hazard at time $t$ given scalar covariates $U_i$ and FPC scores $\xi_{il}$, and $\log h_0(t)$ is the log baseline hazard function. $L$ is often chosen such that the resulting FPC scores cumulatively account for at least 92% of observed variance [35]. The estimated coefficient $\boldsymbol{\theta}^T = (\boldsymbol{\gamma}^T, \boldsymbol{\beta}^T)$ is obtained using the R package mcgv [36].

The following likelihood ratio test (LRT) [37] can then be used to investigate the significant association between the spatial information embedded in the FPC scores $\xi_{il}$ and mortality risk.

$$H_0 : \beta_1 = \ldots = \beta_L = 0 \text{ v.s. } H_a : \beta_j \neq 0 \text{ for at least one j, } 1 \leq j \leq L$$

P-values that are smaller than the significance level $\alpha$ correspond to rejecting the null hypothesis $H_0$ at the $\alpha$ level.

## 2.2 Data

We used datasets from non-small cell lung cancer (NSCLC) and triple-negative breast cancer (TNBC) datasets collected using multiplex immunohistochemistry (mIHC) and multiplexed ion beam imaging (MIBI) platforms, respectively, to evaluate the applicability of our proposed model.

**2.2.1 Non-small cell lung cancer (NSCLC).** Tissue slides collected from 153 patients with non-small cell lung cancer were sequentially stained with antibodies specific for CD19, CD8, CD3, CD14, major histocompatibility complex II (MHCII), cytokeratin, and DAPI. The slides were then imaged on the Vectra 3.0 microscope (Akoya Biosystems). The acquired images were then processed using Akoya's inForm tissue analysis software to obtain a data matrix

with rows corresponding to individual cells and multiple columns corresponding to x- and y-coordinates of each cell on the image, individual marker expressions, and cell phenotypes. Each individual had three to five images corresponding to small regions of the tissue sample. Due to the sparsity in terms of number of cells for some images, we decided to select the image with the maximum number of cells to represent each subject. More details can be found in Johnson et al. [34].

**2.2.2 Triple-negative breast cancer data.** TNBC biopsies were compiled into a tissue microarray (TMA) slides and stained with 36 antibodies targeting regulators of immune activation such as PD1, PD-L1, etc. The slides were imaged using the multiplexed ion beam imaging (MIBI) mass spectrometer. Based on the expression profiles of the 36 markers, each individual cell can be classified. Details about nuclear segmentation and cell phenotyping of the 41 images can be found in [5]. Within this cohort, two patients did not have clinical information available regarding survival outcomes, and one patient's imaging data was corrupted with a high level of noise. As a result, only data on 33 patients were considered in the analysis.

# 3 Real application results

We compared the proposed approach with other diversity measures including Shannon and Simpson indices [38], EDI [28], and Mantel correlation [29] with regard to the significant association between the spatial heterogeneity in the TME and overall survival.

## 3.1 NSCLC

Motivated by a recent study [34] on the of spatial proximity of tumor to immune cells in NSCLC TME, we explored the spatial heterogeneity in various immune cell subsets such as T cells, B cells, and tumor cells. Fig 2A illustrates the distribution of four different immune cell types including CD14+ cells, CD19+ B cells, CD4+ and CD8+ T cells, and CK+ cancer cells, in four representative individuals. Utilizing the entropy measures introduced in Section 2.1.1 for these $I = 5$ categories, the heterogeneity in spatial distributions of these cell types was captured

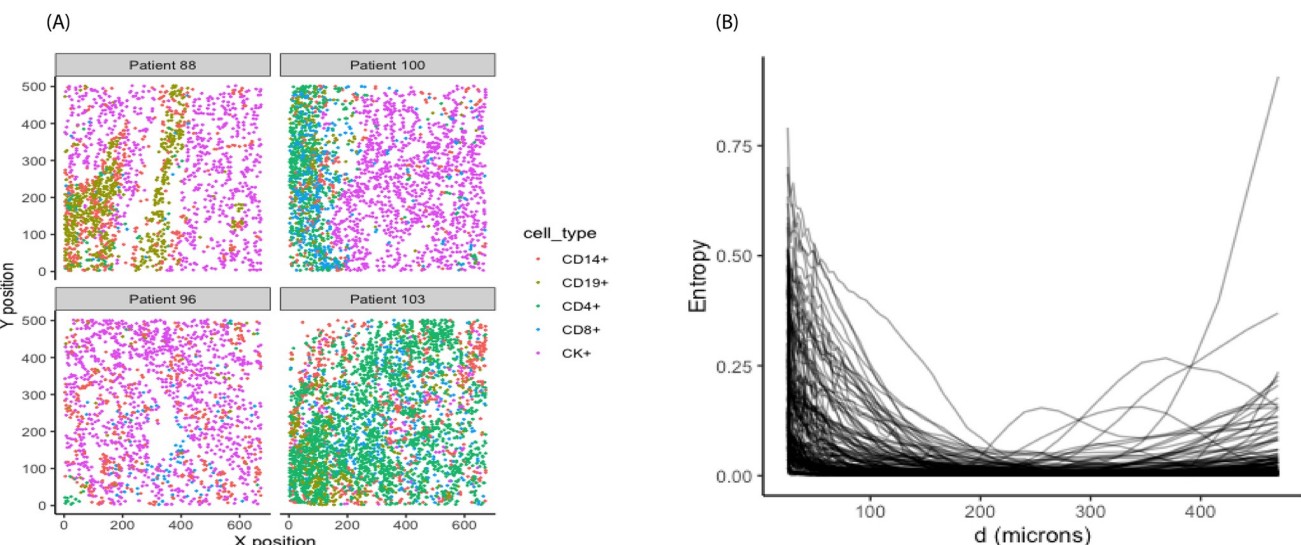

**Fig 2. Lung cancer dataset.** (A) Representative images with distribution of immune cells including CD14+ cells, CD19+ B cells, CD4+ and CD8+ T cells, and CK+ tumor cells. (B) Spatial entropy of the five cell types as a function of inter-cell distances.

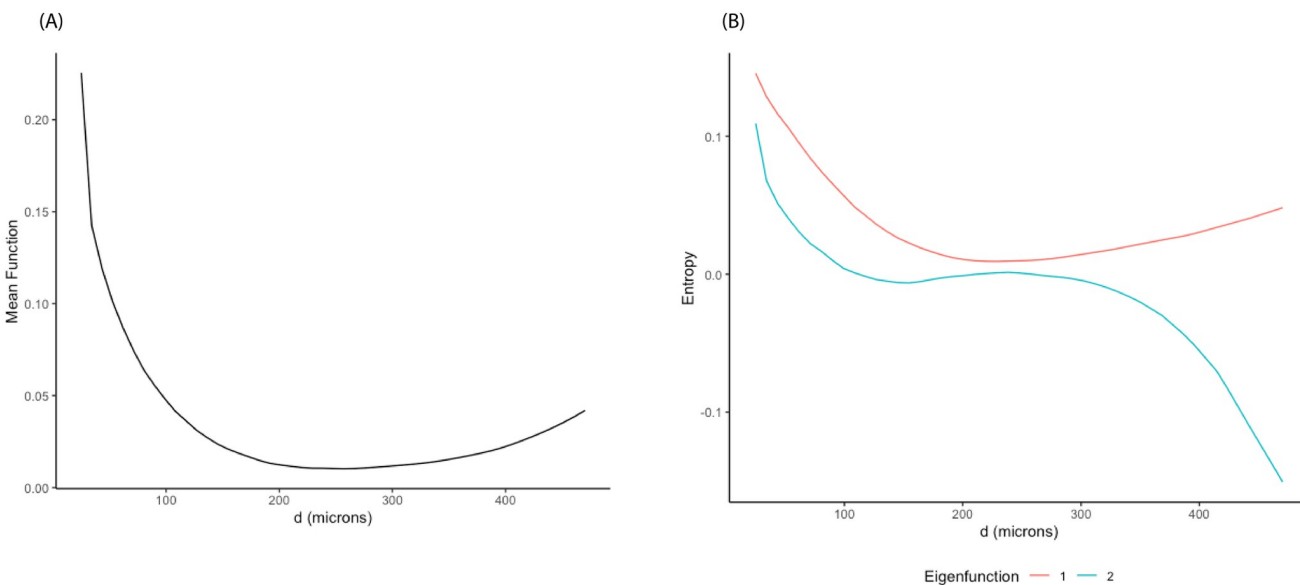

**Fig 3. FPCA results from SPI curves in NSCLC dataset.** (A) Mean function. (B) First two eigenfunctions.

for each individual image. Fig 2B shows spatial entropy measures as a function of inter-cell distances, which we call SPI curves, for all 153 subjects. In particular, there was a high level of variation in spatial entropy values at distances less than 200 $\mu$m at which some individuals expressed high entropy values while others had values close to zero.

The SPI curves were used as input for the FPCA analysis defined in Section 2.1.2 to obtain the estimated FPC scores. Fig 3 shows the estimated mean function (A) and the first two eigenfunctions (B). The estimated mean function reflected the overall trend starting at relatively high entropy values at short distances ($\leq$ 200 $\mu m$), then leveling off close to zero for distances beyond 200$\mu m$. The first eigenfunction showed similar trend to the mean function while the second one expressed a clear contrast in spatial entropy values between short ($\leq$ 200 $\mu m$) and long ($>$200 $\mu m$) distances. The first two FPC scores accounted for 93% of the total variation.

By fitting the two selected FPC scores directly into model (4), we investigated the relationship between spatial heterogeneity in distributions of immune cells in the TME and survival outcome, in addition to subject age. Full and restricted models were fit to test the hypothesis in Section 2.1.3. We obtained p-value of 0.027, indicating a significant non-zero effect of spatial heterogeneity in TME cellular composition on the overall survival at the 5% level of significance.

Section S.2 (S1 Text) summarizes Cox regression model outputs for each of the diversity metrics with additional Kaplan-Meier survival curve for the clustered vs. random GD using the Mantel correlation. None of the metrics identified significant association between spatial heterogeneity in the TME cellular composition and overall survival.

## 3.2 TNBC

Motivated by the mixing scores defined by Keren et al. [5] capturing the level of immune-tumor interactions, we focused on exploring the difference in the spatial distributions of immune and tumor cells in the TME via spatial entropy measures. Fig 4A displays the spatial distributions in four representative images. With the number of categories $I$ = 2, the SPI curves (Section 2.1.1) were computed for 38 subjects as shown in Fig 4B. For short inter-cell distances

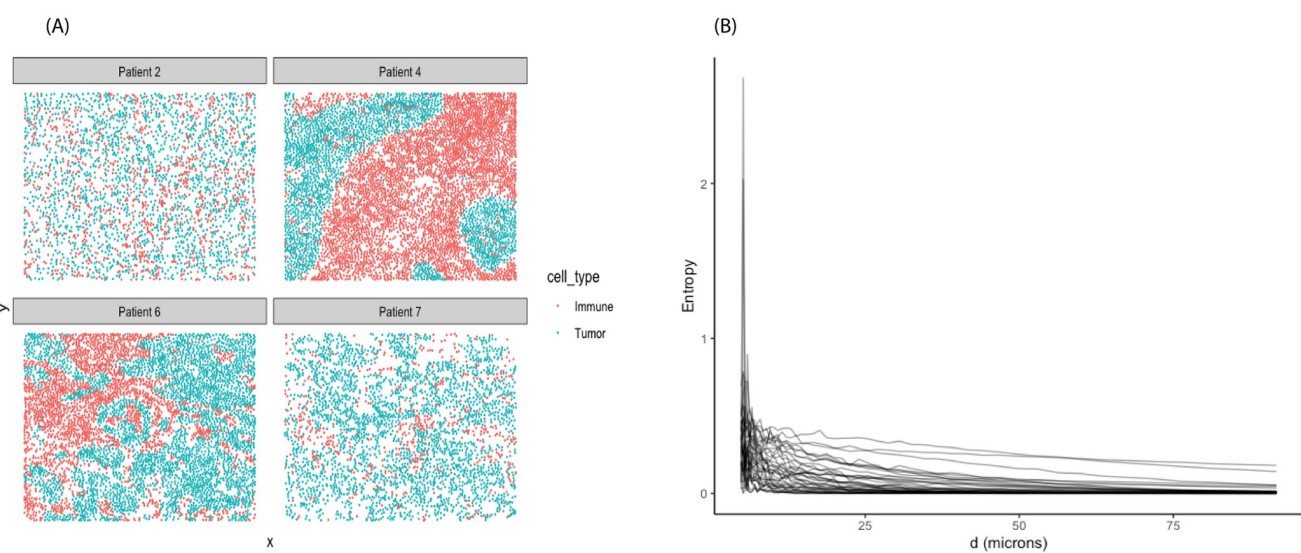

**Fig 4. TNBC dataset.** (A) Representative images with distribution of tumor and immune cells. (B) Spatial entropy of immune and tumor cells as a function of inter-cell distances.

($\leq 30~\mu m$), the spatial entropy values were relatively high compared to those beyond $50~\mu m$. Specifically, high values of spatial entropy at short distances were due to some individuals having small clusters of immune and tumor cells. As the distances increased, cells of different types started scattering more evenly, leading to the spatial entropy dropping close to zero. Similar to Section 3.1, we used FPC analysis on the estimated SPI curves to obtain the corresponding FPC scores. Fig 5A and 5B display the estimated mean function and first three eigenfunctions, respectively. An overall trend of high entropy values at short distances then

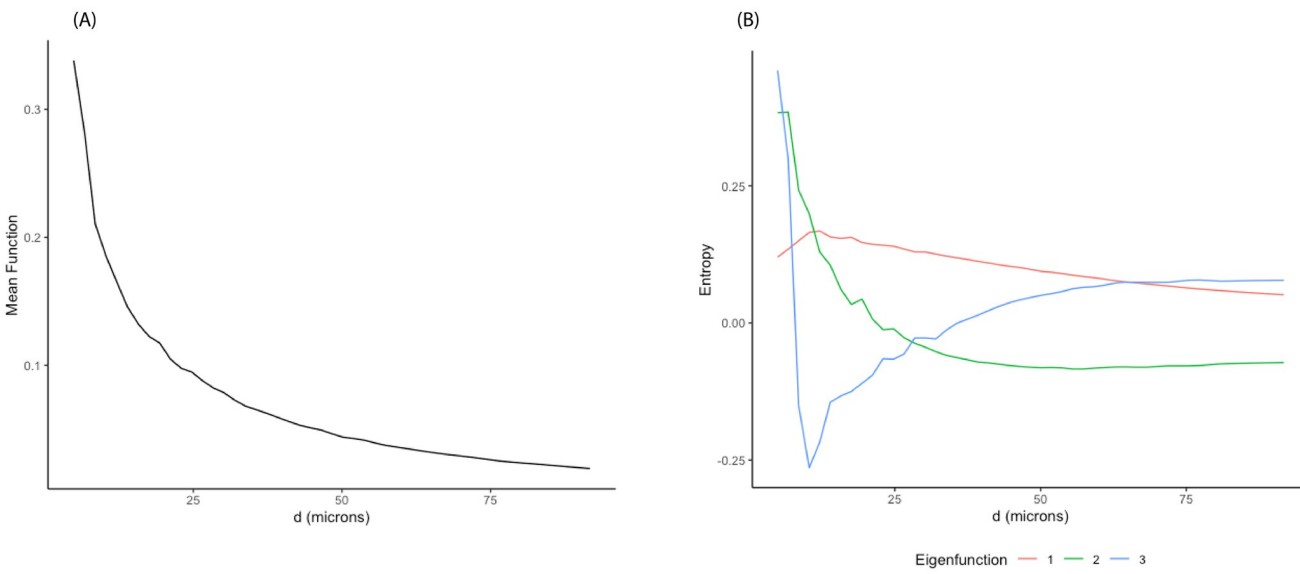

**Fig 5. FPCA results from SPI curves in TNBC dataset.** (A) Mean function. (B) First three eigenfunctions.

dropping off close to zero was reflected in the mean function and the first two eigenfunctions. In addition, the third eigenfunction depicted a contrast in the spatial entropy values between short distances ($\leq 35\ \mu m$) and distances beyond $50\ \mu m$.

Utilizing model (4), we also tested the relationship between spatial heterogeneity in the distributions of the two cell types in the TME of breast tumor samples and mortality risk. In particular, we fitted a full model with the first three FPC scores as predictors in addition to age. A restricted model, on the other hand, only included age as predictor. With a limited sample size ($n = 33$), the LRT of the two models still resulted in a p-value of 0.046 indicating marginally significant association between the heterogeneity in spatial organizations of immune-tumor interactions and survival outcome. Section S.3 (S1 Text) summarizes Cox regression model outputs for each of the diversity metrics with an additional Kaplan-Meier survival curve for the clustered vs. random GD using the Mantel correlation. None of the metrics identified significant impact of spatial heterogeneity in the TME immune-tumor composition and overall survival.

We have conducted additional analyses using two real datasets including ovarian cancer data (as part of the Bioconductor ExperimentHub package VectraPolarisData [39]) and an immunohistochemistry (IHC) data on lung tissues during COVID-19 progression [40]. Specifically, in the ovarian data analysis, we explored the spatial heterogeneity in various immune cell subsets such as T cells, B cells, and tumor-associated macrophages, in relative to CK+ tumor cells by computing spatial entropy. Similar to the analyses above, we also investigated the association between the spatial heterogeneity and survival outcomes. On a different note, for the COVID-19 data, we utilized spatial entropy measures among 17 cell types to classify different stages of COVID-19 progression. More details are provided in Sections S.4 and S.5 of S1 Text.

## 4 Simulation studies

### 4.1 Setup

We performed simulations studies to evaluate the performance of the proposed approach. We considered a dataset of 100 images ($N = 100$). For simplicity, we considered a total of $I = 5$ cell types: CD14+, CD19+, CD4+, CD8+, and CK+ per image. The total number of cells per type was equally fixed at 600, leading to the total cells $n_c = 3000$ per image. For each dataset, we assumed that there were two groups of subjects. The number of subjects per group followed a binomial distribution with a probability of 0.5, i.e., $N_g \sim Binom(N, 0.5)$ for $g = 1, 2$. We considered two spatial configurations: clustered vs. random (Fig 6A and 6B), where subjects in groups 1 and 2 were generated from each configuration, respectively. In other words, the proportion of each cell type stayed consistent across subjects, only their spatial distributions varied. Fig 6C shows the two reference spatial entropy curves corresponding to each configuration. If cells of each type were randomly scattered in a given image, the corresponding spatial entropy values were approximately zero across all distance ranges. Conversely, any configuration deviated from the complete randomness would result in spatial entropy curve above zero. Section S.7.1 of S1 Text provides a sensitivity analysis to explore the impact of rotating a spatial configuration at various angles on the resulting spatial entropy.

Next, subject-specific entropy curves were generated by adding noise to the reference curves. We constructed three scenarios corresponding to the three levels of additive noise: small, medium, and large (Section S.5.1, S1 Text). FPCA was performed on the simulated spatial entropy curves to obtain the estimated FPC scores $\xi_l^*$ (Section 2.1.2). Again, the number of FPCs $L^*$ was chosen such that at least 92% of the total variation was accounted for. Additionally, the scalar covariate $Z_i^*$ was simulated from the normal distribution with mean and

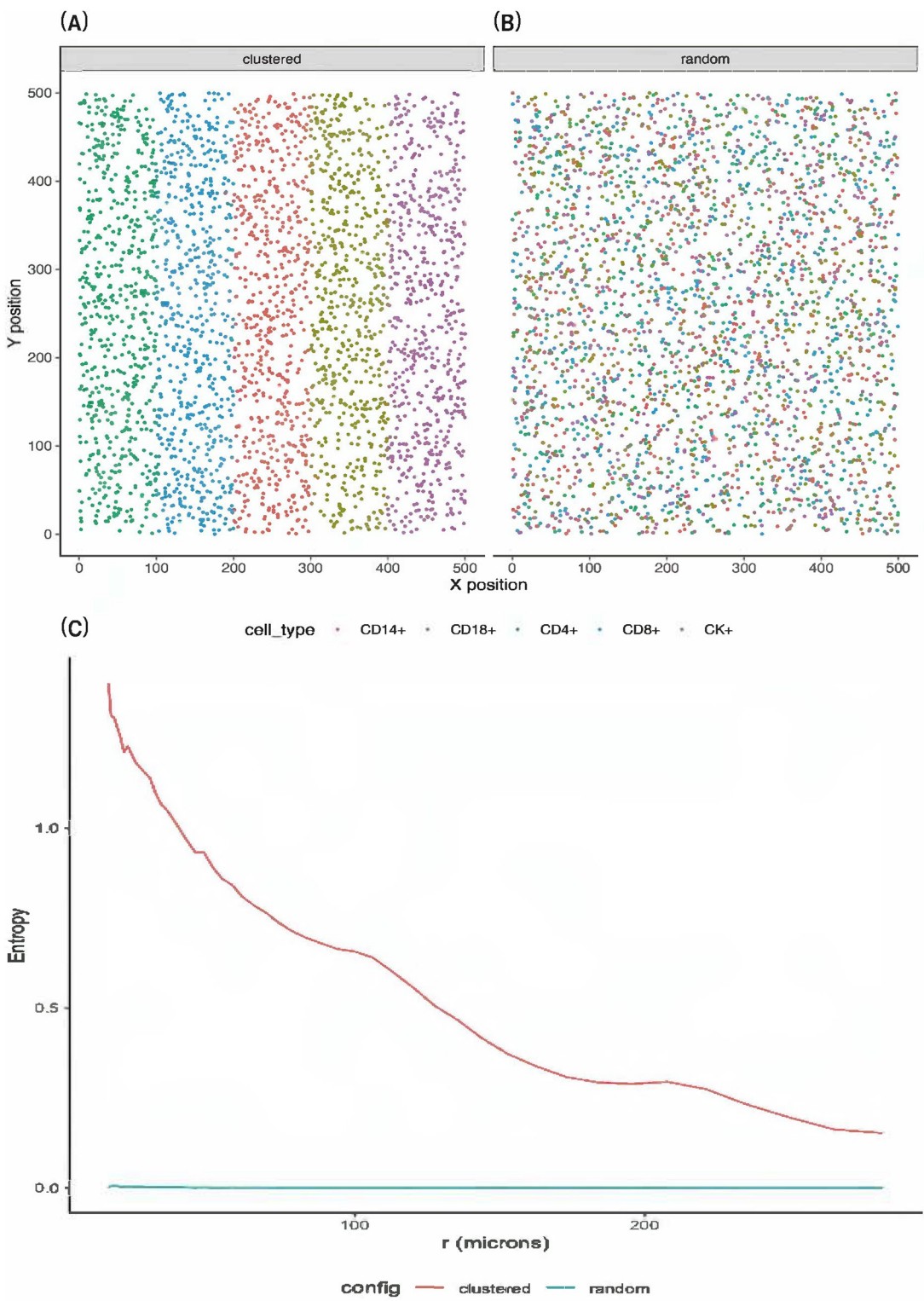

**Fig 6. Simulated spatial configurations.** Two reference spatial configurations: clustered (A) and random (B) of five different cell types: CD14+, CD19+, CD4+, CD8+, and CK+. (C): Corresponding spatial entropy at multiple distance ranges for each configuration.

standard deviation obtained empirically from the distribution of age. After mean centering and unit scaling $Z_i^*$ and $\xi_i^*$, the linear predictor was simulated as $\eta_i^* = Z_i^* + \sum_{l=1}^{L^*} \xi_{il}$ From the fitted model in Section 3, we obtained the estimated cumulative baseline hazard $\int_0^t h_0(x)dx$, which was then used to generate a survival function for each individual, such that $\tilde{S}_i(t) = \exp\left\{-e^{\eta_i^*} \int_0^t \lambda_0(x)dx\right\}$. The estimated survival times $T_i^*$ were generated from the survival function; and the censoring times $C_i^*$ were simulated based on the empirical distribution of the observed censoring times.

## 4.2 Predictive performance

At each level additive noise, four datasets of different sizes (N = 100, 200, 500, and 1000, respectively) were simulated following the procedure in Section 4.1. Each dataset was partitioned into training (75%) and testing (25%) sets. Three models were fit using the training set: (1) a model accounting for both clinical predictor and spatial heterogeneity (2) a model accounting for only spatial heterogeneity, and (3) a model accounting for only clinical predictor. The estimated linear predictor $\hat{\eta}_i^{(u)}$, u = 1, 2, 3 was obtained from the testing set for the $u$th model. At each sample size, normalized root mean squared errors $NRMSE^{(u)}$ was computed as the average of squared differences between the predicted $\hat{\eta}_i^{(u)}$ and the "true" linear predictor $\eta_i^*$ such that $NRMSE^{(u)} = \frac{\sqrt{N_t^{-1} \sum_{i=1}^{N_t} (\hat{\eta}_i^{(u)} - \eta_i^*)^2}}{N_t^{-1} \sum_{i=1}^{N_t} |\eta_i^*|}$, with $N_t$ denoting the number of subjects in the testing set.

We repeated the simulation for 100 iterations and recorded the average NRMSE for each of the three models across four sample sizes N = 100, 200, 500, 1000 in Table 1. Fig 7 displays the distribution of NRMSEs for the three models at each noise-added level across the four sample sizes. Note that when there was a low level of subject-specific variation (Fig 7, first column), the separation between the two spatial configurations was clear (S9A Fig). As a result, all three models performed almost equivalently. The variation of NRMSEs decreased as sample sizes increased from N = 100 to N = 1000. However, when the additive noise was greatly increased, the difference in the spatial entropy curves across the two patterns was not as pronounced. In the extreme case when N = 100, by accounting for the impact of such spatial heterogeneity, models (1) and (2) yielded significantly smaller NRMSEs, as compared to model (3). However, as the sample size increased, the gain in accounting for spatial impact (i.e., models (1) and (2)) in addition to just a clinical predictor (i.e., model (3)) was no longer appreciable.

**Table 1. Normalized root mean squared errors (NRMSE) across three different models, with four sample sizes (N = 100, 200, 500, 1000), and at three levels of additive noise (small, medium, and large).** Corresponding standard deviations are recorded in parentheses.

| Noise level | Model | N = 100 | N = 200 | N = 500 | N = 1000 |
|---|---|---|---|---|---|
| Small | (1) | 0.51 (0.44) | 1.05 (0.03) | 1.08 (0.02) | 1.08 (0.01) |
| | (2) | 0.70 (0.40) | 1.09 (0.02) | 1.09 (0.01) | 1.09 (0.01) |
| | (3) | 1.03 (0.05) | 1.05 (0.02) | 1.08 (0.01) | 1.08 (0.01) |
| Medium | (1) | 0.41 (0.16) | 1.06 (0.03) | 1.09 (0.02) | 1.10 (0.01) |
| | (2) | 0.61 (0.11) | 1.11 (0.03) | 1.11 (0.02) | 1.11 (0.01) |
| | (3) | 1.04 (0.06) | 1.06 (0.03) | 1.09 (0.02) | 1.10 (0.01) |
| Large | (1) | 0.40 (0.20) | 1.06 (0.04) | 1.11 (0.03) | 1.12 (0.02) |
| | (2) | 0.67 (0.09) | 1.12 (0.03) | 1.13 (0.02) | 1.13 (0.02) |
| | (3) | 1.04 (0.07) | 1.08 (0.03) | 1.11 (0.02) | 1.13 (0.01) |

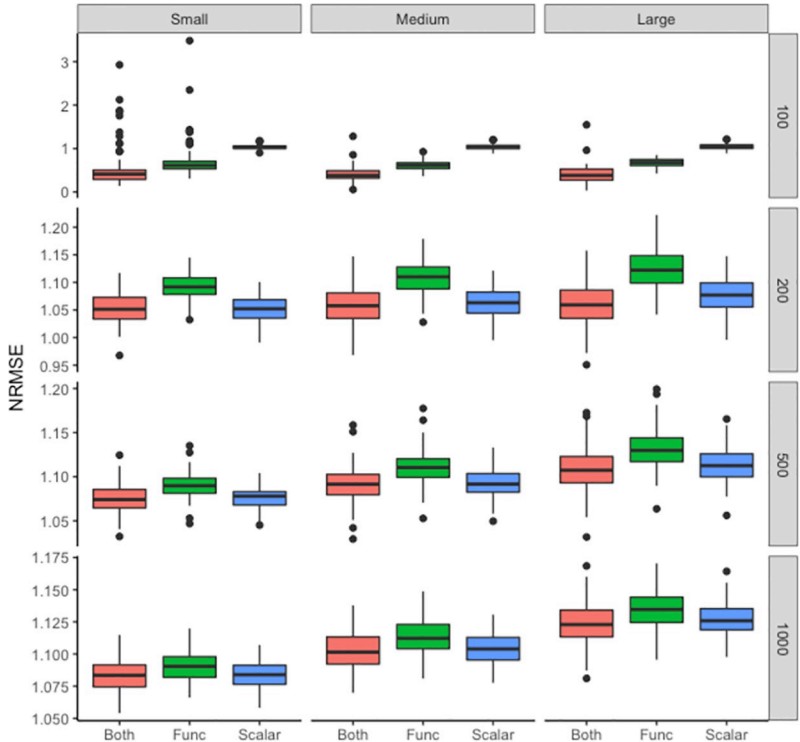

**Fig 7. Normalized root mean squared errors (NRMSE) across three different models.** Distribution of NRMSEs for each of the three models: a model accounting for both clinical predictor and spatial heterogeneity (red), a model accounting for only spatial heterogeneity (green), and a model accounting for only clinical predictor (blue). Columns from left to right represent three levels of additive noise (small, medium, and high). Rows from top to bottom display four sample sizes: N = 100, 200, 500, and 1000. The lower the NRMSE, the better the predictive performance.

Sections S.6.2 and S.6.3 of S1 Text provide two additional simulation studies to specifically assess the performance of the proposed framework compared to the model that incorporates Shannon entropy as a predictor.

## 5 Discussion and conclusions

Spatial organization of cells in the TME plays a critical role in gaining insights into tumor development, progression, and invasion. Recent advances in imaging technology enable investigators to collect single-cell data with an additional dimension of spatial resolution. Conventional metrics typically capture the spatial differential between TME cellular configurations by either exploring the cell distributions in a pairwise fashion or aggregating the heterogeneity in all cell types without considering for their spatial distributions. As a result, neither has fully accounted for the heterogeneity caused by both cellular diversity and spatial patterns of multiple cell categories. Alternatively, we utilize spatial entropy measures to decompose the conventional Shannon entropy into spatial mutual information and residual entropy, which account for the contribution of space and cellular diversity, respectively, to the overall heterogeneity. Then, we apply functional principal component (FPC) analysis for sparse data to the subject-specific spatial entropy trajectories to estimate the FPC scores. The scores are served as predictors in a Cox regression model to investigate the impact of spatial heterogeneity in the TME on survival outcome, in addition to other clinical variables.

Using the NSCLC dataset as a case study, we study the spatial patterns of four different immune cell subsets including CD14+ cells, CD19+ B cells, CD4+ and CD8+ T cells, in relative to CK+ tumor cells across 153 individuals. After fitting the top two FPC scores into the Cox regression model, we find that the spatial heterogeneity in TME immune composition has a significant non-zero effect on the overall survival ($p = 0.027$). The approach is further validated on the TNBC dataset to find the association between the diversity in spatial distributions of tumor cells relative to immune cells and risk of mortality. Given a relatively small sample size, such association is marginally significant. Additionally, through simulation studies under different spatial configurations, we demonstrate that the proposed method has a significantly higher predictive power by accounting for both clinical effect and the impact of spatial heterogeneity, when sample size is small.

In this paper, we utilize the spatial entropy measures to characterize the heterogeneity across distributions of multiple cell types. While the approach does not provide a breakdown of which pairs of cell types driving the spatial heterogeneity due to its aggregated nature, it serves as an initial step in identifying specific distance intervals with significant variation to focus on. Subsequently, existing approaches such as the K-function can be employed to analyze specific pairs of cells of interest individually. Additionally, the accuracy of the spatial entropy relies heavily on the upstream procedure of cell segmentation and phenotyping. In other words, if cells are not segmented and phenotyped correctly, the estimated spatial entropy values would reflect spurious spatial heterogeneity. One possible solution would be to randomly permutate the cell labels to obtain the empirical distribution of the spatial entropy curves. Then, the mean spatial entropy instead of the observed counterpart would then be used as input for the FPC analysis. The selected FPC scores would serve as predictors in the Cox regression model to investigate the association between the spatial heterogeneity of cells and patient overall survival.

## Supporting information

**S1 Text. Supplementary information for "FunSpace: A functional and spatial analytic approach to cell imaging data using entropy measures".**
(PDF)

**S1 Fig. First five functional principal components (FPC) obtained from the NSCLC dataset.** For each FPC, the mean function is overlaid with +/- FPC score multiplying 2 standard deviations of the associated score distribution.
(TIF)

**S2 Fig. Histograms of first four FPC scores obtained from the NSCLC dataset.** FPC scores were obtained by applying FPCA on the spatial entropy curves from the NSCLC dataset. The scores were centered around 0.
(TIF)

**S3 Fig. Kaplan–Meier curves for the overall survival probability from the NSCLC dataset, stratified using the Mantel correlation.** Subjects were classified as clustered vs. random based on the permutation test of the Mantel correlation. P-value of 0.24 indicates non-significant difference in survival probability in two groups.
(TIF)

**S4 Fig. Kaplan–Meier curves for the overall survival probability from the TNBC dataset, stratified using the Mantel correlation.** Subjects were classified as clustered vs. random based on the permutation test of the Mantel correlation. P-value of 0.66 indicates non-significant

difference in survival probability in two groups.
(TIF)

**S5 Fig. Ovarian cancer dataset.** (A) Representative images with distribution of immune cells including CD19+ B cells, CD4+ T cells, CD8+ T cells, CD68+ macrophages, and CK+. (B) Spatial entropy of the five cell types as a function of inter-cell distances.
(TIF)

**S6 Fig. FPCA results from SPI curves in IHC COVID-19 dataset.** (A) Mean function. (B) First three eigenfunctions.
(TIF)

**S7 Fig. IHC COVID-19 dataset.** (A) Representative images with distribution of 17 different cell types (e.g., B cells, CD4 T cells, etc.) (B) Spatial entropy of all cell types as a function of inter-cell distances.
(TIF)

**S8 Fig. FPCA results from SPI curves in ovarian cancer dataset.** (A) Mean function. (B) First two eigenfunctions.
(TIF)

**S9 Fig. Simulation scenarios.** (A) low additive noise. (B) medium additive noise. (C) large additive noise. Three levels of noise were added to the reference SPI curves (clustered vs. random) to generate subject-specific SPI curves.
(TIF)

**S10 Fig. Representative simulated cell distributions for additional simulation 2.** The number of cells for each cell type was simulated from a negative binomial distribution with the mean and dispersion parameter randomly selected from two ranges [200:500] and [1:3], respectively. The x-coordinates of some randomly selected cell were varied to create overlapping between clusters of cell types.
(TIF)

**S11 Fig. Configuration rotation.** The simulated clustered point pattern (i.e., original) introduced in Fig 6A was rotated at various angles including 180˚, 90˚, 60˚, 45˚, and 36˚.
(TIF)

**S12 Fig. SPI under different configuration rotations.** At each rotation, pairwise distances between all cells were calculated based on their corresponding x- and y-coordinates. Distance ranges were subsequently computed. Finally, the spatial entropy curve across all distance ranges was obtained.
(TIF)

**S13 Fig. SPI under $K$ different distance ranges.** $K$ values were incremented from 25 to 100 in steps of 5. At each $K$, a sequence of distance breaks was generated by linearly decreasing from $d_K$ to 0 on a log scale. Distance ranges were subsequently computed. Finally, the spatial entropy curve across all distance ranges was obtained.
(TIF)

**S14 Fig. Empirical distribution of $p(z_r|w_k)$.** One representative image in the NSCLC dataset was used for demonstration. Based on pairwise distances between cells, we generated $K = 50$ distance breaks $w_k$ for $k = 1, \ldots, K$. At each $w_k$, the co-occurrences between cell types $p(z_r|w_k)$ were obtained. We then built this panel of histograms using the relative frequencies of the cell

type co-occurrences.
(TIF)

**S15 Fig. Illustration for SPI calculation.** A point pattern consisting of 30 cells, with 10 cells per each type A, B, and C, was simulated. Circles of radius 0.25 were drawn around each point to identify co-occurrences of cell types with the first distance range $w_1 = (0, 0.25]$. Specifically, there were 1 AA, 1 BB, 3 CC, 0 AB, 0 AC, and 0 BC.
(TIF)

## Acknowledgments

We thank the Human Immune Monitoring Shared Resource and support of the University of Colorado Human Immunology and Immunotherapy Initiative for their expert assistance in multiplex IHC and generation of the non-small cell lung cancer (NSCLC) and ovarian cancer datasets.

## Author Contributions

**Conceptualization:** Thao Vu, Debashis Ghosh.

**Data curation:** Thao Vu, Souvik Seal, Tusharkanti Ghosh, Mansooreh Ahmadian, Julia Wrobel, Debashis Ghosh.

**Formal analysis:** Thao Vu, Julia Wrobel, Debashis Ghosh.

**Funding acquisition:** Debashis Ghosh.

**Investigation:** Thao Vu, Debashis Ghosh.

**Methodology:** Thao Vu, Souvik Seal, Julia Wrobel, Debashis Ghosh.

**Project administration:** Debashis Ghosh.

**Resources:** Julia Wrobel, Debashis Ghosh.

**Software:** Thao Vu.

**Supervision:** Debashis Ghosh.

**Validation:** Thao Vu, Debashis Ghosh.

**Visualization:** Thao Vu.

**Writing – original draft:** Thao Vu.

**Writing – review & editing:** Thao Vu, Souvik Seal, Tusharkanti Ghosh, Mansooreh Ahmadian, Julia Wrobel, Debashis Ghosh.

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
