## [Decision Letter · Decision Letter 0]

29 Mar 2023

Dear Dr. Vu,

Thank you very much for submitting your manuscript "FunSpace: A functional and spatial analytic approach to cell imaging data using entropy measures" for consideration at PLOS Computational Biology.

As with all papers reviewed by the journal, your manuscript was reviewed by members of the editorial board and by several independent reviewers. In light of the reviews (below this email), we would like to invite the resubmission of a significantly-revised version that takes into account the reviewers' comments.

We cannot make any decision about publication until we have seen the revised manuscript and your response to the reviewers' comments. Your revised manuscript is also likely to be sent to reviewers for further evaluation.

Sincerely,

Ellis Patrick

Guest Editor

PLOS Computational Biology

Lucy Houghton

Staff

PLOS Computational Biology

The reviewers have raised a number of concerns about your work as it stands. All of these appear to be addressable. I encourage you to respond to all of the comments below and submit a revision your manuscript for further consideration. Further to this,

1) I strongly recommend that you make functions to run FUNSPACE available on CRAN or Bioconductor to ensure it can be used easily by the community.

2) Given the description on the github page, I would request that you clarify that this work is not concurrently under consideration at Bioinformatics.

3) Your figures have rendered skewed and of varying quality in the submission process. You should double check this.

Regards,

Ellis Patrick

Reviewer's Responses to Questions

**Comments to the Authors:**

Reviewer #1: Vu et al. present FunSpace, a novel approach to leverage spatial entropy measures at multiple distances to account for the spatial heterogeneity of cellular architectures. The authors applied FunSpace to both simulated and real datasets and demonstrated that the feature extracted by FunSpace is informative in predicting survival for lung and breast cancer patients. While the paper is clearly written, several concerns dampened our overall enthusiasm.

Major concerns

1. One of the major claims in the paper is that the spatial entropy measures proposed in this manuscript can better describe the cellular architectures than other metrics. The author only demonstrated this point theoretically. However, no comparison was conducted in the downstream simulations or real data applications. Such comparasion is necessary to tell if the spatial entropy is better than other existing measures in practice. For example, in the application to the NSCLC dataset, it is not surprising to find that the spatial heterogeneity described by the spatial entropy has a significant association with overall survival. I wonder if we use other metrics, e.g., the Shannon entropy, instead of the spatial entropy, and perform the same downstream analysis, we will arrive at the same conclusion.

More comprehensive comparisons are desired to systematically show the advantage of spatial entropy over others. In the simulation and real data analysis, the authors can replace the spatial entropy with Shannon entropy, conduct the rest analysis, and compare the results with the current ones achieved using spatial entropy.

2. Although the author claims that FunSpace can be used to study various tumor types, the current evaluation is only based on lung and breast cancer data. It is unclear how generalizable this method is. It appears various tuning and adjustments of some parameters are needed. The idea may work well for lung and breast cancer data, but may not be readily usable in other cancer types.

3. Following my previous comments, some critical parameters' values are determined arbitrarily, which limits the usefulness of FunSpace in practice. For example, the distance breaks dk are critical in the calculation of spatial entropy. The author states that they can “be flexibly chosen depending on the specific applications” (line 117). How the choice of these parameters affects the downstream results? How to choose the value of these parameters when analyzing different tumor data? Some guidance is in demand.

4. The spatial entropy summarizes the cell type heterogeneity of the tumor region using only one value. It is an oversimplified representation of the complex tumor microenvironment, which have very limited interpretability. In both the application to lung and breast cancer datasets, the final conclusions are both “the association between the heterogeneity in spatial architectures of immune-tumor interactions and survival outcome is significant”. It would be great if the author could provide more insight into this association. For example, which cell pair’s distribution is more important in this association?

Minor concerns

1. Some improvements can be made to make the method section easier to read. For example, in Lin 116, the variable K is used without definition. In line 145, lowercase s is not defined.

2. Since spatial entropy is a novel measure, it would be helpful to discuss its properties. For example, will different spatial configurations yield the same spatial entropy? Is the spatial entropy invariant under rotation?

Hopefully, these questions/suggestions can help to further strengthen this manuscript.

Reviewer #2: In this manuscript, the authors claimed that the spatial relationship of cell types in tumor micro-environment (TME) has a significant impact on patient’s survival outcome. They proposed to use spatial entropy to measure the spatial relationship of cell types in TME, and proposed to utilize the functional principal component analysis (FPCA) of the spatial entropy as covariates in the Cox Model in estimating the survival of subjects.

In terms of the proposed method, I have the following comments:

1. In interpreting equation (1), the authors mentioned “In our context, the maximum entropy is reached when cells of different types are scattered evenly on a given image. “. I believe the authors meant same number of cells in every cell type? Because the in this equation the spatial information is not included

2. Is x_i a vector, and the element is 1 where cell is in cell type i and 0s everywhere else? Or is it an integer indicating the cell type of a cell (like 1, 2, 3, …)

3. The definition of co-occurrence of cells is not fully clear to me - is it the possible combination of types of two cells (x_i and x_i’), or is it defined as the co-occurrence of two cell types within some region? I assume that it is the combination of cell types of any two cells in the image?

4. Can the authors justify the decomposition of H(Z)? And the definition if SPI(Z)? Is it different from SPI_k(Z)?

5. What is the empirical distribution of p(zr|wk) in a real image?

6. Can the authors justify using local smooth functions for SPI and give detailed steps of smoothing? Especially for X(s) and G

I’d like to point out that the notation is a bit hard to follow -

1. i is cell type in method section but subject in model section

2. same for the X notation, cell identification in method section, but a function in model section

3. and same for Sik

Reviewer #3: The review is uploaded as an attachment.

**Have the authors made all data and (if applicable) computational code underlying the findings in their manuscript fully available?**

Reviewer #1: None

Reviewer #2: Yes

Reviewer #3: Yes

PLOS authors have the option to publish the peer review history of their article (what does this mean?). If published, this will include your full peer review and any attached files.

Reviewer #1: No

Reviewer #2: No

Reviewer #3: No
---

## [Decision Letter · Decision Letter 1]

4 Sep 2023

Dear Dr. Vu,

We are pleased to inform you that your manuscript 'FunSpace: A functional and spatial analytic approach to cell imaging data using entropy measures' has been provisionally accepted for publication in PLOS Computational Biology.

Best regards,

Ellis Patrick

Guest Editor

PLOS Computational Biology

Lucy Houghton

%CORR_ED_EDITOR_ROLE%

PLOS Computational Biology

The reviewers are all satisfied that you have addressed their concerns. Thank you again for contributing this spatial analysis approach to the community.

Reviewer's Responses to Questions

**Comments to the Authors:**

Reviewer #1: All comments have been addressed satisfactorily

Reviewer #3: Thank the authors for their efforts in the revisoin and I don't have further questions.

**Have the authors made all data and (if applicable) computational code underlying the findings in their manuscript fully available?**

Reviewer #1: None

Reviewer #3: Yes

PLOS authors have the option to publish the peer review history of their article (what does this mean?). If published, this will include your full peer review and any attached files.

Reviewer #1: No

Reviewer #3: No

---

## [Editor Report · Acceptance letter]

15 Sep 2023

PCOMPBIOL-D-22-01701R1 

FunSpace: A functional and spatial analytic approach to cell imaging data using entropy measures

Dear Dr Vu,

I am pleased to inform you that your manuscript has been formally accepted for publication in PLOS Computational Biology. Your manuscript is now with our production department and you will be notified of the publication date in due course.

With kind regards,

Zsuzsanna Gémesi
